# Seismic Vulnerability of Segmental Bridges with Drop-In Span by Pushover Analysis

Piero Colajanni * , Michele Fabio Granata and Lidia La Mendola

Department of Engineering, University of Palermo, Viale delle Scienze, 90128 Palermo, Italy;
michelefabio.granata@unipa.it (M.F.G.); lidia.lamendola@unipa.it (L.L.M.)
* Correspondence: piero.colajanni@unipa.it; Tel.: +39-091-238-96550

**Featured Application: When evaluating the seismic vulnerability by means of non-linear static analysis of bridges characterized by the presence of drop-in spans, attention must be paid to the choice of the force distribution, which must be able to excite the structure in such a way as to represent the oscillations of all its parts and the possible phase deviation angle.**

**Abstract:** Insight into the application of pushover analysis to prestressed concrete segmental bridges built in the 1950s–1970s by cantilevering with medium-large span length is provided. Seismic assessment must be carried out considering the whole structural response and, in particular, the task of tall piers, bearings, and drop-in spans with Gerber saddles, which are likely to be subjected to girder pounding and/or unseating. In this paper, the assessment of seismic vulnerability is initially performed by linear modal dynamic analysis; then, the efficiency in assessing the seismic response of different methods of pushover analyses is compared, assuming as a benchmark the results of non-linear time history analysis. The outcomes show that, for the bridge with the drop-in span, criteria for selecting the load pattern considered in pushover analysis, the reliable modeling of the bearings, and tall piers play a dominant role in the assessment of the seismic vulnerability, particularly in longitudinal motion.

**Keywords:** bridge seismic vulnerability; pushover analysis; load pattern; bearing modelling; longitudinal response; pounding; unseating

## 1. Introduction

Prestressed concrete segmental bridges built in the 1950s–1970s by cantilevering with medium-large span lengths are now at the end of their nominal service life, even though they are currently in service. Hence, road authorities need to carry out assessments of the static and seismic vulnerabilities of these bridges. The latter needs an estimate of the internal force redistribution allowed by the plastic resources of the structure and a comparison of resistance, deformation capacity, and demand, ruled by the collapse mechanism that is activated. In this context, nonlinear response history analysis (NRHA) is the most trustworthy procedure to find out the strength and ductility demand of bridges due to seismic action. Nevertheless, by carrying out NRHA, large computational efforts are required in either reliable modeling of the bridge and seismic excitation, the execution of the analysis, and results post-processing to get design values of the response quantities needed for the design. Furthermore, the great sensitivity of the response to seismic excitation modeling and the non-linear cyclic behavior of the bridge have deferred the use of NRHA by the designer. Conversely, non-linear static analysis (NSA), often referred to as PushOver Analysis (POA), has become, in the last decades, the most common analysis method for the seismic vulnerability assessment of buildings and, more recently, of bridges.

Many studies in the literature, e.g., [1–5], explored the extension of POA, initially formulated for the evaluation of the nonlinear seismic response of buildings, to the assessment

of the seismic vulnerability of bridges. Most of the papers highlighted that the extension is not straightforward since bridge dynamic behavior very often depends on a large number of modes, while POA is particularly efficient when the seismic response is mainly ruled by the fundamental mode of vibration.

To solve this weakness, several multimodal procedures derived for the assessment of the nonlinear seismic response of buildings [1,5] were extended to the assessment of bridge seismic response; in other cases, procedures specifically formulated for bridges [3–8] have been attempted. Recently, machine learning methods were also used for the seismic risk assessment of bridges [9].

Regarding the assessment of the seismic vulnerability of buildings, Modal Pushover Analysis (MPA), formulated and accurately developed by Chopra and Goel [6,7], is one of the most effective procedures. The method can be seen as a direct extension to the nonlinear behavior of the well-known dynamic modal analysis with response spectrum technique for the elastic system since it is based on the following: pushover analyses performed independently for each relevant mode with invariant load pattern; evaluation for each mode of the target displacement by definition of an equivalent Single Degree Of Freedom (SDOF) system; evaluation of strength and deformation demand according to the corresponding configuration of the structure derived by each pushover analysis; and subsequent superimposition of the displacement response parameters by an appropriate modal combination rule (SRSS or CQC). A more conventional approach for taking into account the effect of higher modes in pushover analysis is to derive load patterns, either fixed [4] or time-variant (adaptive) [5], on the basis of the elastic, initial, or tangent modal response of the structure.

Concerning the adaptation of MPA to bridges, Kappos et al. [3] highlighted the following items: the modes to be considered should be properly restricted since the number of modes needed to obtain a total modal participant mass ratio of 85% could be very large; for each mode, an appropriate control point should be chosen; in this connection, Kappos stressed that the natural choice for the control point is the deck mass center, or the top of the nearest pier, the point of the deck with the maximum displacement, or the center of the seismic forces for the given mode. This choice has a relevant influence on the evaluation of the target displacement, and on the influence of the modes on the response when entering the nonlinear range and the bridge deformation is no longer proportional to the elastic eigenvector; the assessment of target displacement should be performed with special attention in order to guarantee that the response of each mode is consistent to the design intensity of the seismic action; to this aim, the pushover curve should be idealized by the equivalent bi- or three-linear curve on suitably chosen criteria; the use of nonlinear displacement spectrum is advised; an appropriate choice of the modal combination rule adopted for superposition of the modal response (SRSS o CQC) is required.

In [2], it has been observed that the response in the transverse direction is the most sensitive to higher modes. Concerning the longitudinal response, pounding or unseating can affect the bridge behavior. The collapses of bridges and/or large-scale damages induced by pounding have been reported for heavy earthquakes (1989 Loma Prieta [10,11], 1994 Northridge [12], 1995 Kobe [13], 2010 Chile [14], 2011 Christchurch [15]), highlighting that in many circumstances, inadequate control of longitudinal response is among the main causes of exceeding service and ultimate conditions [16,17]. Pounding and unseating occur between the deck and abutment or, due to out-of-phase vibrations, in the decks between adjacent segments in multi-span bridges. Out-of-phase vibrations can lead to two different causes of collapse: if the two decks move away from each other to a considerable extent, unseating may occur; conversely, if two adjacent girders move closer together, pounding can occur. It has been widely recognized that either the overcoming of serviceability conditions, unseating, or failure due to pounding action in multi-span or framed bridges is dependent on the movement range of expansion joints. The main aspects are the actual gap between the abutment and deck, the restraint stiffness, the frame stiffness ratios, the earthquake loading with or without asynchronous excitation at the supports, and the yield

strength [18–20]. Damages due to pounding have a strong impact on post-earthquake restoration activities, leading to long periods of out-of-service or the requirement of large costs for recommissioning.

More in detail, in [20], it was found that a great significance of the pounding/unseating risk only occurs when the two adjacent decks have significantly different vibration periods. In [19], it was found that, when the response is limited to the elastic range, the pounding tends to reduce the bending moment on the piers. Still limiting the analysis to the response in the elastic field of bridges, Bi et al. [21] studied the influence of the dynamic characteristics of the bridge and the frequency content of the input on the minimum distance of separation that avoids the pounding of contiguous decks. They used the Tajmi and Kanai models, considering the asynchronous motion at the supports as a result of the spatial variation of the motion. They found that the minimum gap required to avoid the pounding is not only determined by the frequency ratio of the contiguous frames but also by the ratio between their frequency and the frequency of the seismic excitation. In these cases, the need to extend the analysis to the nonlinear behavior of the bridge has been widely recognized [22,23]. In [16], the risk of pounding in bridges with piers of different heights and thus different vibration periods is analyzed through the inelastic response range, proving that the ratio of the periods of vibration and the gap size between the deck and abutment are the major relevant parameters that influence the risk of pounding. By decreasing the ratio between the minimum and maximum periods of vibration of the two adjacent frames, the effect of the characteristics of the seismic input is found to be less relevant in determining the risk of pounding and unseating. Concerning the risk of pounding between adjacent buildings, Miari and Jankowsy [24] observed that the relationship provided in the literature for the minimum gap able to avoid pounding is unreliable when soil-structure interaction is considered. They revised the current models for establishing the seismic gap of buildings in order to eliminate structural collision, starting from the well-known modal combination rules for the evaluation of the maximum expected displacements for each deck, evaluated through the response spectrum technique. In particular, they refer to the absolute sum (ABS) used in the Uniform Building Code of 1997 [25], the Square Root of the Sum of Squares (SRSS) [26], the Complete Quadratic Combination (CQC), where correlation coefficients for white noise modeling of the seismic excitation are used [27], and the double equation difference (DDC) [28], where the aforementioned correlation coefficients are affected by the signum minus. They used the expression of combination coefficients as a function of the period ratio and difference proposed by Naderpour et al. [29]. Based on extensive numerical analyses, Miari and Jankowsy [24] proposed new equations for different types of soil as a function of the minimum period and the period ratio of the two systems.

POA is able to provide a consistent quantitative assessment of the risk of unseating or pounding phenomena at the interface between the deck and the abutments. However, it is an unsuitable method to analyze the actual behavior, either before or after unseating or pounding phenomena, between vibrating end regions of different parts of the structure, where two-sided impact can occur. Thus, the estimation of the risk of pounding or unseating at the ends of the drop-in span is challenging since this behavior is mainly ruled by the out-of-phase oscillation [18]. The evaluation of the tails of the probability function of overcoming a barrier of relative displacements between two non-linear oscillators with close periods of vibrations due to seismic excitation is a very complex issue, making unreliable the evaluation of a given fractile of the relative displacements. Thus, only statistical information given by the correlation coefficients can model the average value of the risk of overcoming a given limit state [30]. To this aim, linearization techniques [31] are often unable to provide reliable results. Thus, standard POA, in a general case, can assess a lower bound only of the seismic acceleration for which unseating or pounding phenomena can occur, assuming that the oscillations are in phase opposition. When the periods of vibration of the two adjacent frames are almost equal, and thus in-phase oscillations are expected, a reliable estimation of the risk of unseating or pounding at the abutments can be provided.

In this research, the application of both modal dynamic analysis and pushover techniques to the assessment of the seismic vulnerability of bridges are conducted, highlighting that they must be applied with special care. Moreover, insight is provided for the application of POA as a tool for the preliminary evaluation of the structural deficiencies to be filled with seismic strengthening interventions of different natures [32–34]. The seismic vulnerability in the longitudinal and transverse directions is investigated for a framed bridge with a clamped deck to the pier and for a continuous box girder disconnected from piers by comparing the modal dynamic analysis. Then, for a case study belonging to the most vulnerable typology, the evaluation of the response by modal dynamic analysis is performed, highlighting the role of combination coefficients in the modal combination rule. Then, the effectiveness of several procedures for pushover analyses is investigated, assuming as a benchmark the response assessed by non-linear time history analysis.

## 2. Segmental Bridges Built by Cantilevering

The girder bridges with medium-large span lengths and variable box cross-sections are built mainly through cantilevering technology, that is, segments that are assembled one to the other until they complete half of the span length through a cantilever scheme from the pier to the midspan. Afterward, when the two cantilevers meet at the midspan, the span length is closed. The evolution of these bridges took place at the same time as the development of prestressed concrete, which allowed their development since the introduction of upper prestressing, segment by segment, made possible the equilibrium of the forces involved with the increasing length of the cantilever. This method of construction affects both the overall static scheme (especially in the ratio of the central and side spans), the layout of the prestressing tendons, and the overall stress state at the end of construction [35].

The first bridges built with this technology in Europe and Italy had the cantilevers clamped at the top of the pier, and this constraint remained effective both during construction and in service life. The cantilever technology was put into operation using segments cast in situ on formworks fixed to the adjacent segment, already cast. In this way, both the prestressing reinforcements and the ordinary ones could have continuity in the joints between one segment and the next one. Although this technology guarantees complete continuity of the cantilever during construction and the possibility of having mild reinforcements in the joints that can face any tensile stress or cracking, on the other hand, it led to significant downward deformation over time due to creep. This was due to the age of casting and the hardening of concrete on site that was loaded with its self-weight from the beginning, at an early age. Another characteristic of the first cantilevered concrete bridges was that they adopted a prestressing technology made of rigid bars (originally patented by Dywidag), which, although made adherent to concrete and joined together, have shown reduced performance compared to the modern flexible tendons made of strands. Additionally, they show greater stress loss both instantaneously during the stressing phase and later, over time. In the static scheme, the segmental bridges of the first generation keep the clamp between the deck and pier and a hinge at the center of the span (Figure 1a), at the junction between the two cantilevers, or even a Drop-In Span (DIS) of small length that finds a place between the two tips of the cantilevers through Gerber saddles. Such disconnection at the center of the span maintains the isostatic behavior, or at least a low degree of redundancy, and does not activate the frame effect that occurs in bridges where the deck is continuous and clamped to each pier. In the 1960s and 1970s of the 20th century, this scheme was preferred both because it was not very sensitive to ground settlements and because it maintained the same structural behavior between construction and service life, simplifying the prestressing layout. The effects of creep and prestressing bars associated with in situ casting led engineers to almost abandon this solution, which was used in bridges of this generation, even of large spans, till the 1970s.

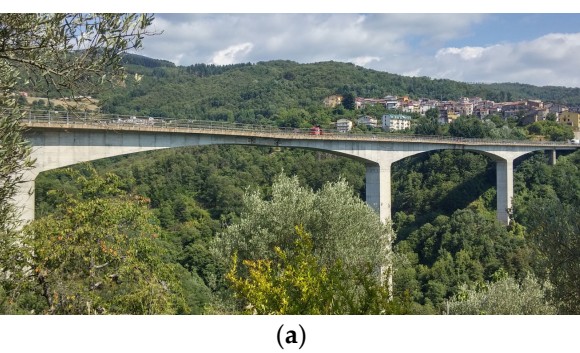
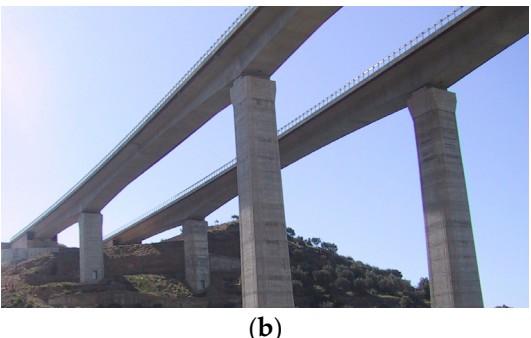

(**a**)                                                                                     (**b**)

**Figure 1.** (**a**) Cannavino bridge on the roadway between Paola and Cosenza, Italy, with clamped deck on piers and drop-in span; (**b**) Bridge on the A20 motorway in Sicily, Italy, with continuous box girder on piers.

The evolution of these bridges has led engineers to prefer a final scheme in service life in which the deck is a continuous beam on piers that are disconnected from the deck girder after the cantilever construction when the central span is closed. In this scheme, which is always redundant, the clamp between the pier and deck is temporary for the phase of construction only; hence, it ensures the equilibrium of the cantilever, but at the end of construction, the continuous beam behavior occurs (Figure 1b). In this configuration, the upper prestressing tendons already inserted in the construction stage remain in service life too, but other prestressing tendons are added in the bottom slab of the boxed segments at the midspan, which are necessary to balance the positive moments due to service loads.

In fewer cases, multi-span bridges were built, closing completely the deck in the midspan until reaching the multiple-frame scheme. This scheme, although it presents a significant natural contribution of axial force (arch-frame effect), on the other hand, increases the degree of hyperstaticity of the system, complicating the solutions necessary to absorb the deformations and stresses due to temperature as well as the introduction of hyperstatic prestressing. The behavior of this last typology in the seismic action is significantly different. Hence, in the following, only the cases of framed bridges with midspan disconnection and continuous bridges will be initially treated, which are more common.

### 3. Dynamic Modal Analysis (DMA) with Response Spectrum Technique

In this section, as a first approach, the seismic response is evaluated by the linear dynamic modal analysis with response spectrum for segmental bridges with the two different schemes described above, namely the framed one and the continuous girder. The behaviors of these two typologies are investigated through the analysis of the responses of two testing bridges having the same geometrical dimensions and mechanical properties. The responses are compared to recognize which scheme is more sensitive to the seismic action and what the critical aspects are for the two schemes. All the analyses are carried out on a one-dimensional FE model of the structures. The Standard Complete Quadratic Combination (CQC) rule is applied for modal contribution combinations.

The analyzed structural scheme consists of a three-span bridge with a total length of 245 m: the central span is 115 m long, the side ones are 65 m, and pier height is 49 m; the deck is composed of a one-cell prestressed concrete box girder of 9.50 m wide with web thickness of 0.4 m; the top slab has constant thickness of 0.25 m, while the bottom slab has variable thickness from 0.80 m on piers to 0.20 m at mid-span. The total height of the cross-section is variable too, from 7 m over the piers up to 2.2 m at the midspan and near the abutments. Materials are concrete grade C40/50 (characteristic cylinder strength $f_{ck}$ = 40 MPa), mild reinforcement steel with characteristic yielding stress $f_{yk}$ = 450 MPa, and prestressing steel of Dywidag bars with $f_{pyk}$ = 1000 MPa. The pier section is a hollow section with dimensions of 450 × 510 cm, a thickness of 0.40 m, and a total height of 43 m between the foundation and the deck, the pier cross-section has 0.8% longitudinal reinforcement and stirrups of diameter $\phi$14 mm with 4 legs and 0.20 m spacing, characterized by a M-N interaction

resistance domain in which, for the value of axial force due to permanent loads, 66.5 MN, the correspondent bending moment in the domain is 847 MNm. Soil structure interaction is neglected to highlight the structural behavior only. Thus, the piers are clamped at the base, and the abutment restraints will be particularized depending on the structural scheme analyzed. Furthermore, the hypothesis of equal seismic action at the foot of the piers was made, considering that there is a limited distance between them and the spatial variability of the motion has no influence.

Besides the self-weight of the structure, a superimposed dead load of 3 kN/m$^2$ is considered, resulting in a total seismic weight of 37,127 kN. The elastic response spectrum corresponding to subsoil class "B" of Eurocode 8 [36], with a peak ground acceleration of 0.435 g, is considered for the seismic input characterization.

### 3.1. Analysis of Bridges with Framed Scheme

In this section, the bridge with the deck clamped to the pier is analyzed. The central span is characterized by the presence of a drop-in span (DIS), 19 m long, that finds a place between the two tips of the cantilever through Gerber saddles. Such disconnection at the center of the span maintains the isostatic behavior of the structure. Thus, the connection between the cantilever and the drop-in span is modeled as a fixed hinge on the right side, while on the left side is a roller that prevents transverse and vertical translations only. The latter also represents the restraint at the abutments. The vibration periods of the first mode in longitudinal and transverse directions are 1.70 s and 1.62 s, respectively.

In Figure 2a, the bending moment diagram for the seismic input in the longitudinal direction is represented, resulting in a flexural moment at the pier base and at the top section of 220 MNm and 160 MNm, respectively, on the deck section at the connection with the pier of 335 MNm, and at the side midspan section of 70 MNm. In Figure 2b, the bending moment diagram for the seismic input in the transverse direction is represented, resulting in a flexural moment at the pier bases of 205 MNm and on the deck section at the side span that varies from 55 MNm at the midspan to 30 MNm over the pier, while at the central midspan of 88 MNm.

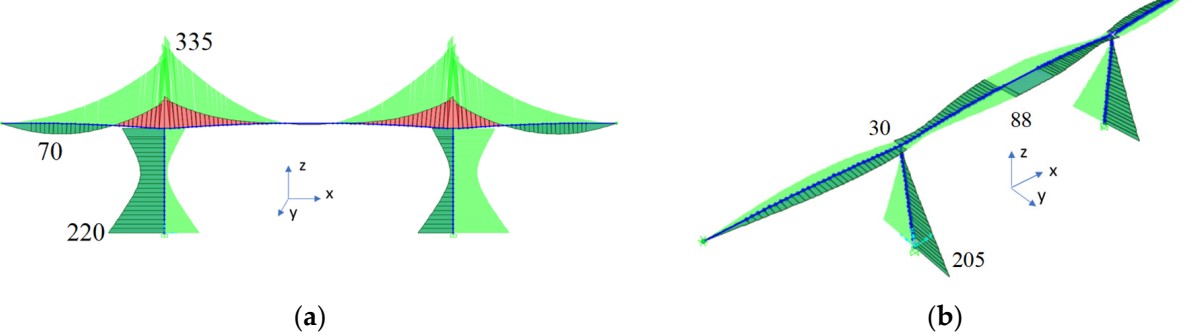

(**a**)　　　　　　　　　　　　　　　　　(**b**)

**Figure 2.** Bending moment on the framed bridge for the seismic action: (**a**) moment $M_y$ for seismic action in longitudinal direction x; (**b**) moment $M_x$ for seismic action in transverse direction y.

### 3.2. Analysis of Bridges with Continuous Beam

In this section, the bridge with the continuous girder is analyzed, characterized by continuity in the central span and one-direction rollers at the pier-deck connections. The restraints at the left abutment differ from the previous model since the translations in the three directions and the rotation along the vertical axis are restrained (fixed abutment). The vibration periods of the first modes in longitudinal and transverse directions are 1.31 s and 1.81 s, respectively.

In Figure 3a, the bending moment diagram for the seismic input in the longitudinal direction is represented, resulting in a flexural moment at the pier base of 122 MNm, on the deck section at the pier connection of 295 MNm, and at the central midspan of 56 MNm. In Figure 3b, the bending moment diagram for the seismic input in the transverse direction is represented, resulting in a flexural moment at the left pier base of 150 MNm, on the right

pier base of 210 MNm, on the deck section at the left abutment of 120 MNm, and at the central midspan of 100 MNm.

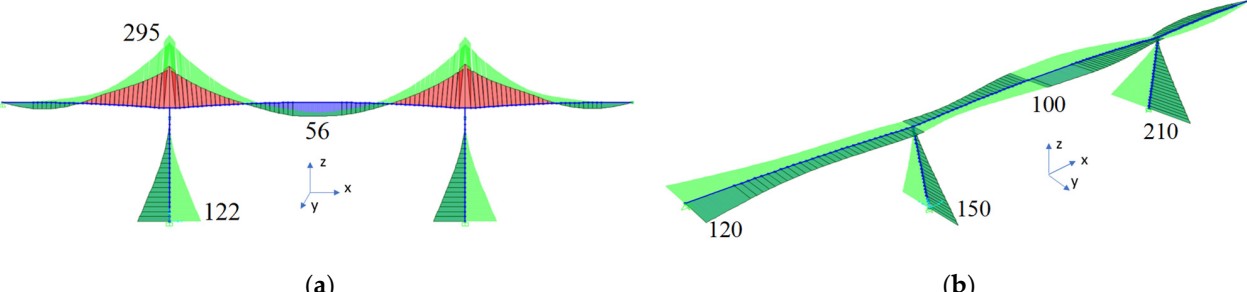

**Figure 3.** Bending moment on the bridge with continuous girder for gravity load and seismic actions (**a**) moment $M_y$ for seismic action in longitudinal direction x; (**b**) moment $M_x$ for seismic action in transverse direction y.

### 3.3. Comparison of Results

The comparison of the results obtained with the seismic input in the longitudinal direction shows that in the framed scheme, the bending moment at the pier base is about 100 MNm larger than in the continuous girder, while at the top pier section, there is not any significant variation. By contrast, in the midspan section, the bending moment is larger for the continuous girder. Regarding the seismic response due to the transverse direction, in this case, the piers of the framed scheme are more heavily loaded. More specifically, due to the asymmetry of the restraints, the right pier is subjected to a bending moment 50 MNm larger than its counterpart in the continuous girder. By contrast, an opposite trend is found on the deck, since in the continuous girder there is a bending moment in the central span of 20 MNm larger than its counterpart in the framed scheme and of 120 MNm at the end of the left span that is vanishing in the framed scheme.

The comparison shows that the framed scheme is more vulnerable in the longitudinal direction due to the circumstance that the deck is clamped to the piers and the abutment restraints do not contribute to withstand seismic action in the longitudinal direction; thus, it should be fully carried on by the piers only.

Based on the previous results, in the following section, the behavior of the framed scheme is analyzed through the pushover analysis, performed with more accurate modeling of the restraints, and emphasizing the response in the longitudinal direction.

## 4. Case Study: The Grottalunga Viaduct

### 4.1. Bridge Structural Scheme and FEM Model

The structure chosen for the case study is the Grottalunga viaduct, designed by Carlo Cestelli Guidi in 1962 and located on the A3-Salerno-Reggio Calabria highway, where the seismic action is represented by the elastic response spectrum corresponding to subsoil class "B" of Eurocode 8 [36], with a peak ground acceleration of 0.333 g.

The viaduct is a framed girder extending approximately 221 m in length and 49 m in height, built by segments through cantilevering technology. It is composed of two side spans of 53 m and a central span of 115 m (Figure 4a). A drop-in span of 19 m length finds a place between the two tips of the cantilevers through Gerber saddles.

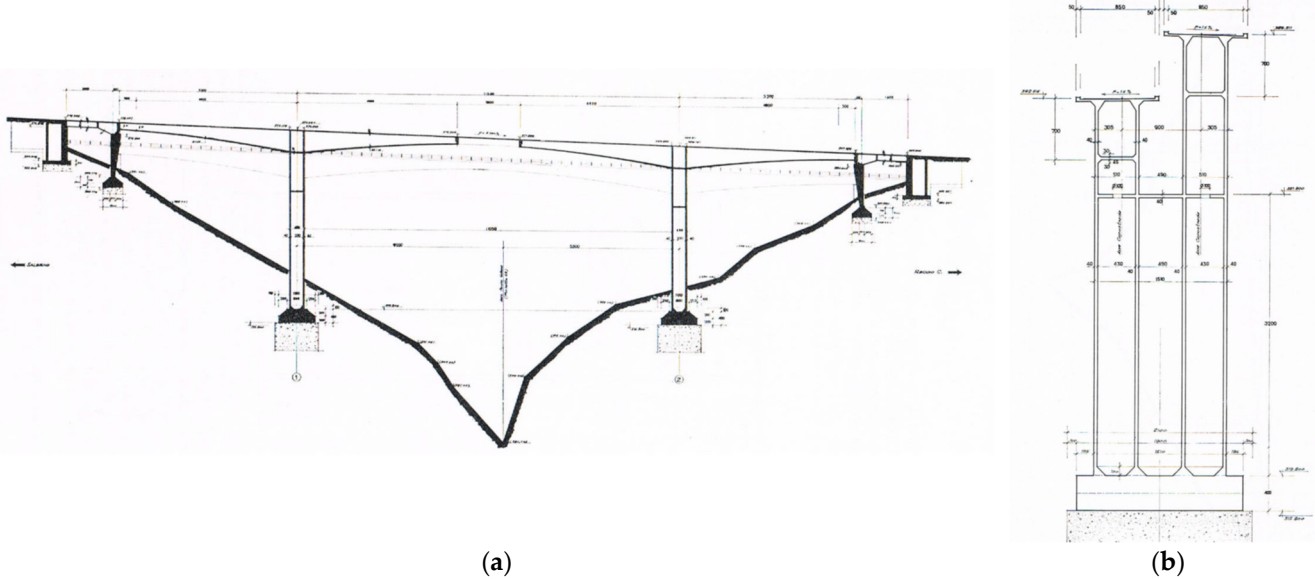

| (a) | (b) |

**Figure 4.** The Grottalunga viaduct; (**a**) longitudinal scheme; (**b**) pier transverse section.

For local needs, the abutments were designed as frames; this led to the following study considering the abutments as fixed points without considering the approach spans but focusing only on the three main spans of the viaduct.

The cantilevered spans are made from prestressed reinforced concrete segments 3.5 m long, with the exception of the segments corresponding to the piers, which measure 2.25 m. The section of each segment (Figure 4b) is box-shaped with 0.4 m thick webs, 0.25 m top slab, and a variable bottom slab, with section heights ranging from 7 m in the connection section to the piers up to 2.2 m at the ends.

The concrete grade is C40/50, and steel bars have a characteristic yielding strength of $f_{yk}$ = 450 MPa. The piers have the same cross-section and reinforcement as those described in Section 3 of this paper; nonlinear behavior was modeled through fiber elements, adopting the Mander model for the confined concrete [37] and the Menegotto-Pinto model [38] for the steel of the reinforcing bars. A more refined model of the restraints was developed. More precisely, in order to model the vertical and torsional resistance of the bearing and the chance of uplifting, a two-dimensional restraint was modeled by rigid elements. In Figure 5a, the restraint on abutments is shown, while in Figure 5b, the connection between the tip of the cantilever and the drop-in span is shown. Both models have the length of rigid elements equal to the width of the deck. For each deck end, two elastomeric pot bearing types are considered, one having the strength to transverse force of 500 KN and the other allowing transverse displacement, both having a vertical tensile strength of 800 kN and free longitudinal displacement in the range of ±100 mm. Similar bearings are considered at the Left end of the Drop-in Span (LEDIS), while at the right end, each bearing restrains the longitudinal displacement with the strength of 500 kN each. Thus, for the longitudinal and transverse directions, the operational limit of performance of the restraint that allows movements (Figure 5d) is $d_{OLS}$ = ±100 mm. Once $d_{OLS}$ is overcome, the deck is placed directly on the abutment or the Gerber saddle, and then an additional displacement of ±50 mm is permitted with a horizontal passive reaction force due to the friction of concrete elements (deck cross-section, saddle, or abutment). The friction coefficient $\mu_c$ = 0.25 was assumed, thus a displacement at the damage limit state $d_{DLS}$ = ±150 mm is assumed. When the $d_{DLS}$ is overcome, either the pounding or the unseating can occur. For restraints that prevent displacements, the yielding displacement of $d_y$ = ± 1 mm and a subsequent plastic displacement of $d_p$ = ±25 mm are modeled, and then the same limits for the unrestrained displacement are assumed (Figure 5e): $d_{OLS}$ = ±100 mm and $d_{DLS}$ = ±150 mm. In order to take into account the effects of the construction stages, initially the dead load was

applied to the bridge, namely the self-weight of the piers and girders and the weight of the deck and superstructure. Then, the restraint reaction was evaluated together with the displacements of the deformed configuration reported in Figure 5c. The left cantilever showed a displacement on the pier top of 43 mm and the right cantilever of −42 mm.

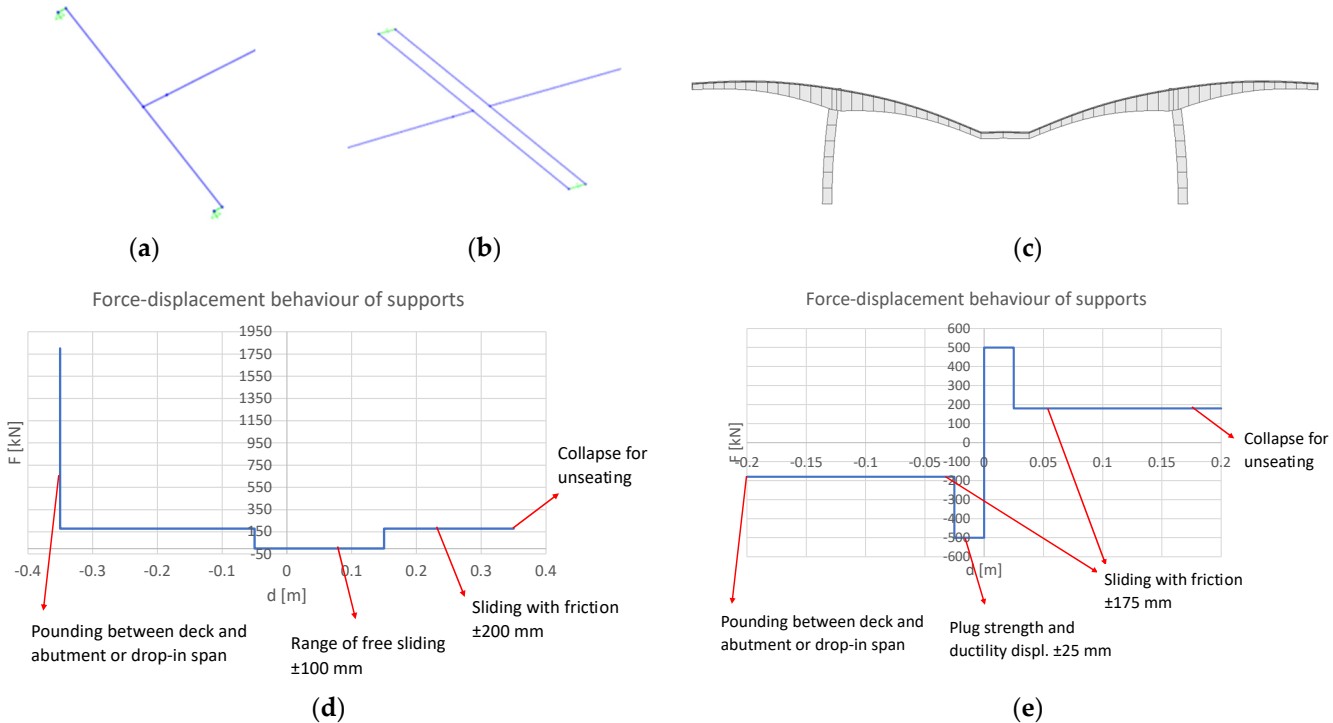

**Figure 5.** Simplified modeling of the bearing: modeling scheme (**a**) at the abutments; (**b**) at the deck-drop-in span connections; (**c**) deformed shape due to dead load (horizontal displacement of +43 mm and −42 mm for the left and right cantilevers, respectively); force-displacement relationship for support with (**d**) unrestrained displacement; (**e**) restrained displacement.

The setting of the nonlinear joint movement range was adjusted to consider that the restraints at the abutments and the connection between the cantilever and drop-in span are placed after the displacement due to the dead load, that is after the construction is completed. Thus, the force-displacement relationship for the support at the left abutment or at the joint between the right end of the left cantilever (RELC) and the left end of the drop-in span (LEDIS), where the longitudinal displacements are enabled, assumes the form represented in Figure 5d. The support where the relative displacements are restrained (as the joint between the left end of the right cantilever and the right end of the drop-in span) maintains the form depicted in Figure 5e.

In Figure 6, periods, modal shapes, and mass participant ratios for the translational directions are depicted, showing that in order to capture the translational response, six modes should be taken into account. Regarding longitudinal behavior, the drop-in span and the constraints divide the structure into two substructures. Remarkably, fifth mode and sixth mode are characterized by the pier deformation shape with an inflection point along the height; they have to be taken into consideration in order to correctly estimate the bridge longitudinal behavior.

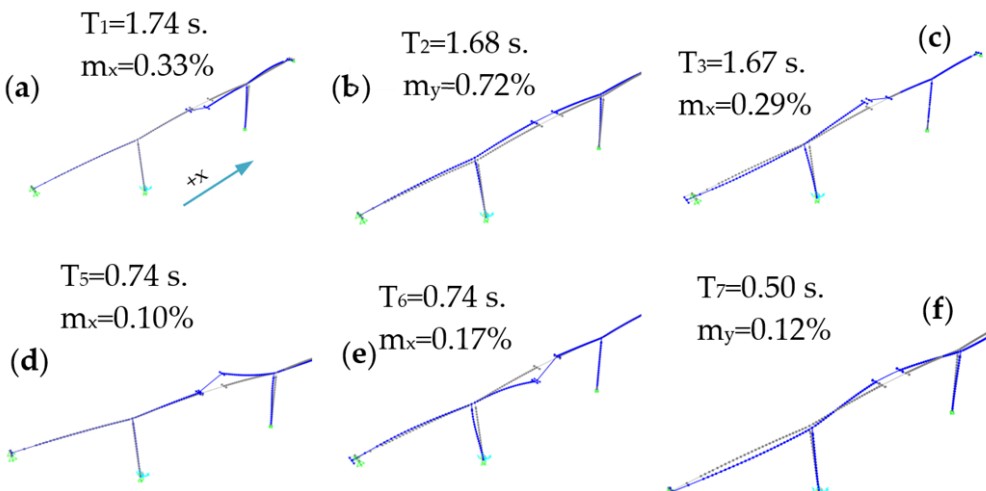

**Figure 6.** Periods, modal shapes, and mass participant ratios. (**a**) first period (**b**) second period (**c**) third period (**d**) fifth period (**e**) sixth period (**f**) seventh period.

### *4.2. Analysis of the Seismic Response in the Transverse Direction*

#### 4.2.1. Modal Dynamic Analysis (MDA)

In the transverse direction, the large mass participation ratio of the 2nd mode, namely $m_y(2) = 0.72$, proves that a single mode is sufficient to represent the response of the structure with good accuracy. By contrast, the subsequent mode that has a significant mass participant ratio, namely the seventh mode, has a mass participant ratio in the transverse direction $m_y(7) = 0.12$ and a period of vibration $T_7 = 0.5$ s. This suggests that, since the two periods of vibration are well separated, whatever modal combination rule is used to assess the expected response, namely SRSS or CQC rule, the results will be almost coincident. Since the analysis is linear, the transverse displacement at the support is restrained, and the resulting deformed shape and the contour of the displacement (mm) for PGA = 0.333 g are shown in Figure 7. The contour of displacement [mm] in the transverse direction was assessed by MDA for ag = 0.33 g. The resulting maximum displacement at the midspan of the bridge is $d_{y,max,CQC} = 283$ mm, and at the top of the pier $d_{y,TP,CQC} = 167$ mm when CQC is used while using SRSS, a corresponding displacement of $d_{y,max,SRSS} = 281$ mm and $d_{y,TP,SRSS} = 166$ mm are found.

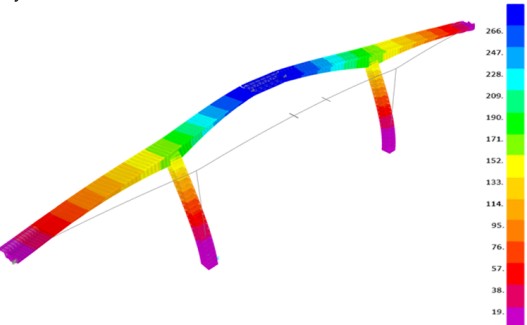

**Figure 7.** Contour of displacement [mm] in the transverse direction assessed by MDA for ag = 0.33 g.

#### 4.2.2. Single-Run Single-Mode Pushover Analysis in Transverse Direction

For the pushover analysis in the transverse direction, the control node was chosen on the left side of the joint between the deck of the left cantilever and the drop-in span. The pushover curve (with the origin assumed in the deformed configuration due to gravitational loads) obtained through a load pattern proportional to the product of the mass times the mode shape for 2nd mode (positive displacement) is depicted in Figure 8a, together with the bi-linearized curve obtained according to the Italian code [39]. When the 2nd mode load pattern is assumed, the first event is the plastic hinge activation (PHA) at the base

of the left and right piers (LRBP) for a displacement of the control node $d_{2,1}$ = 142 mm and a Peak Ground Acceleration $PGA_{2,1}$ = 0.167 g. Afterward, the Yielding of the Plug (YP) on the restraints on the left and right abutments (LRA) occurs for $d_{2,2}$ = 214 mm and a $PGA_{2,2}$ = 0.252 g. A plastic hinge collapse (PHC) for attainment of the maximum rotation at the hinges at left and right pier bases (LRPB)s is reached for $d_{2,C}$ = 236 mm and a $PGA_{2,c}$ = 0.277 g, smaller than the PGA design value. These results are summarized in Table 1.

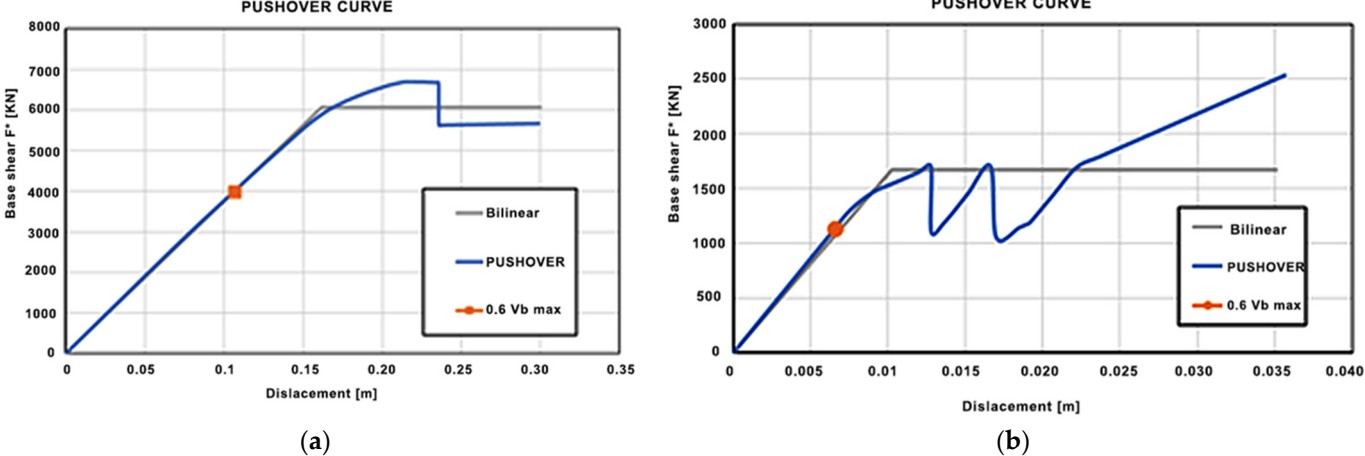

**Figure 8.** Pushover curves for modal load pattern in the transverse direction (**a**) second mode; (**b**) seventh mode.

**Table 1.** Events and related displacement and PGA for PO with mode 2 and mode 7 load pattern, respectively.

| | Second Mode | | | Seventh Mode | |
|---|---|---|---|---|---|
| **Event** | **Displ. (mm)** | **PGA (g)** | **Event** | **Displ. (mm)** | **PGA (g)** |
| PHA LRPB | 142 | 0.167 | YP LRA | 10 | 0.099 |
| YP LRA | 214 | 0.252 | PF RA | 13 | 0.121 |
| PHC LRPB | 236 | 0.277 | PF LA | 17 | 0.157 |
| | | | ES LRA | 19 | 0.181 |

### 4.2.3. Single-Run Multimodal (MM) Pushover Analysis in Transverse Direction

Considering that the second mode in the transverse direction, namely the seventh mode, has a mass participation ratio of 12%, a slightly more reliable estimation of the seismic response is expected with a load pattern consistent with the reaction at the external restraints provided by modal analysis (base of piers and abutments). To this aim, a load pattern in the form $f = M(\phi_1 + \alpha\phi_{12})$ is assumed, where $f$ is the vector of the seismic force, $M$ is the mass matrix of the structure, $\phi_i$ is the $i$-th mode shape. The coefficient of combination $\alpha$ is evaluated by imposing that the ratio of the transverse reaction at the base of the pier and at the abutment obtained by adopting the force distribution $f$ in the elastic phase is equal to its counterpart obtained by linear dynamic modal analyses. The procedure resembles what is prescribed by the seismic code for buildings, where the load pattern is required to be consistent with the story shear obtained by the linear modal analysis.

In Figure 9a, the PO curve is shown, characterized by the yielding of the plug-in at both the left and right abutment restrained sides for a displacement of the midspan of the bridge of $d_{MM,1}$ = 99 mm and a $PGA_{MM,1}$ = 0.129 g. Afterward, the failure of the two plug-in for a displacement $d_{MM,2}$ = 115 mm, and a $PGA_{MM,2}$ = 0.149 g is reached. This is suddenly followed by the activation of the plastic hinges at the bases of both piers for a

displacement $d_{MM,3}$ = 116 mm, and a $PGA_{MM,2}$ = 0.151 g. The last event is the failure of the aforementioned plastic hinges (attainment of plastic rotation capacity) for a displacement $d_{MM,c}$ = 190 mm, and a $PGA_{MM,2}$ = 0.247 g. Comparison of these results with those obtained through the load pattern according to the 2nd mode shows that the contribution due to mode 7 leads to premature yielding and subsequent failure of the transverse restraints at the abutments for lower values of PGA.

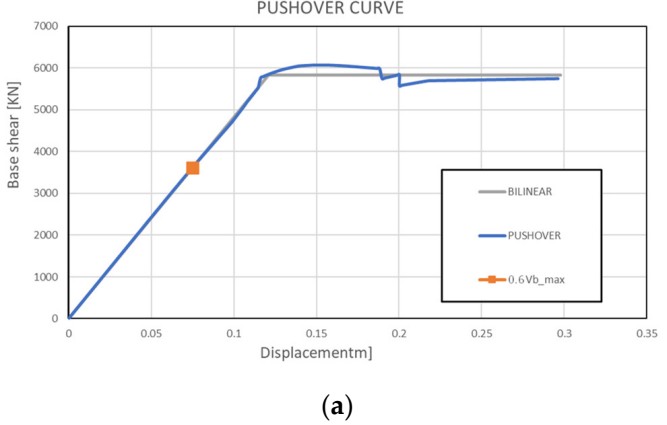

| Event | Displ (mm) | PGA (g) |
|---|---|---|
| YP_LRA | 99 | 0.129 |
| FP_LRA | 115 | 0.149 |
| PHA_LRPB | 116 | 0.151 |
| PHF_LRPB | 190 | 0.247 |

(**a**)

(**b**)

**Figure 9.** Single-run PO analysis with multimodal load pattern: (**a**) PO curve; (**b**) events and related displacement and peak ground acceleration.

4.2.4. Modal Pushover Analysis (MPA) in Transverse Direction

When the seismic response is not strongly non-linear, the procedure that usually provides the better estimation of the actual nonlinear response obtained via nonlinear time history analysis is the MPA proposed by Chopra and Goel [6,7]. This multi-run PO procedure requires that pushover analyses are performed independently for each relevant mode with an invariant load pattern. Thus, the pushover curve for the seventh mode is evaluated and shown in Figure 8b, and the results are summarized in Table 1. The first event is the yielding of the plugs of the restraints on the left and right abutments that occur for $d_{7,1}$ = 10 mm and $PGA_{7,1}$ = 0.099 g. The second one is the failure of the plug of the right abutment, which occurs for $d_{7,2}$ = 13 mm and a $PGA_{7,2}$ = 0.121 g; then the failure of the plug on the left abutment ($d_{7,3}$ = 17 mm and a $PGA_{7,3}$ = 0.157 g) is achieved, and lastly the exit of the supports on both the left and right abutments from their seats, with the beginning of further sliding with friction ($d_{7,4}$ = 19 mm and a $PGA_{7,4}$ = 0.181 g).

In Figure 10a–c, the deformed shape obtained by the single-run pushover analysis according to the second mode and seventh mode is compared with the associated deformed shape obtained via MPA for three different levels of PGA. They are PGA = 0.099 g, which represents the value corresponding to the end of the elastic behavior, PGA = 0.274 g, for which the hinges at the base of the piers exhaust their plastic rotation capacity; and the design PGA = 0.333 g. From the figures, it can be recognized that the displacement demand at the end of the drop-in span and the piers can be evaluated by the 2nd mode single-run pushover analysis, and thus the difference with MPA in assessing the collapse acceleration is less than 2%, while the displacement demand at the abutment supports estimated with MPA is 47% larger for the collapse PGA value.

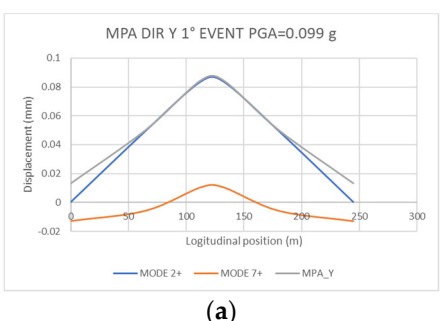

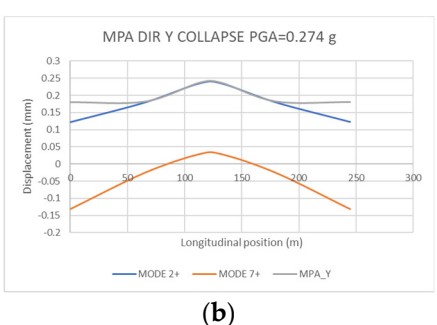

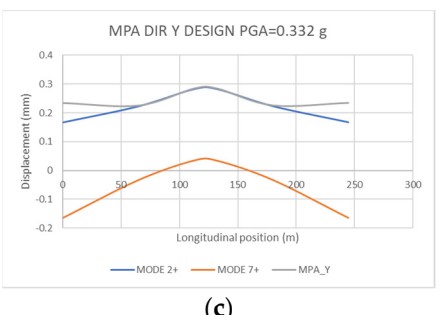

**(a)**  **(b)**  **(c)**

**Figure 10.** Comparison of deformed shapes obtained by single-mode and modal Pushover analysis for PGA equal to (**a**) 0.099 g; (**b**) 0.274 g; (**c**) 0.332 g.

*4.3. Preliminary Discussion for Developing Modal Dynamic Analysis and POA in Longitudinal Direction*

Since the bridge is divided into two substructures by the constraints on the Gerber saddles, two problems arise, depending on the type of analysis that is performed: if MDA is employed, in order to evaluate the risk of pounding or unseating at the connection of the DIS where relative displacement with the cantilever is allowed, a suitable modal combination rule should be utilized; if pushover analysis is performed since single-mode load pattern is able to simulate the oscillation of half structure only, multi-modal load pattern or pushover procedure should be used. In both cases, critical issues are related to the possible occurrence of oscillation between the two cantilevers with a phase shift.

Regarding MDA, when the risk of pounding is evaluated, the use of the absolute sum (ABS) for the combination of the maxima of modal response was soon found to be highly conservative. By contrast, the use of the SRSS rule can be on the unsafe side, especially when the two decks have periods close to each other. In this circumstance, the use of CQC with correlation coefficients derived for white noise representation of seismic excitation is the reference method proposed by the code. The relative displacement $\Delta U$ has to be evaluated as follows:

$$\Delta U = \sqrt{\Delta U_1^2 + \Delta U_2^2 + 2\rho_{12}\Delta U_1\Delta U_2} \tag{1}$$

$$\rho_{12} = \frac{8\zeta^2\beta_{12}^{\frac{3}{2}}}{(1+\beta_{12})\left[(1+\beta_{12})^2 + 4\zeta^2\beta_{i12}\right]} \tag{2}$$

where $\beta_{12} = T_1/T_2 = \omega_2/\omega_1$ while $T_1$, $T_2$, and $\zeta$ are the periods of vibration and the common damping ratio of the two frames.

Cacciola et al. [40] show that, if spectrum-compatible seismic input is considered, a correlation coefficient consistent with the power spectral density PSD should be used in Equation (1), which can be evaluated as follows:

$$\rho_{12,PSD} = \frac{1}{\kappa_{12}}\left(\gamma_{12}\omega_2^2\sqrt{\frac{\lambda_{0,u_2}}{\lambda_{0,u_1}}} + \gamma_{ji}\omega_i^2\sqrt{\frac{\lambda_{0,u_1}}{\lambda_{0,u_2}}} + \varepsilon_{12}\frac{\lambda_{2,u_2}}{\sqrt{\lambda_{0,u_1}\lambda_{0,u_2}}} + \varepsilon_{21}\frac{\lambda_{2,u_1}}{\sqrt{\lambda_{0,u_1}\lambda_{0,u_2}}}\right) \tag{3}$$

$$\gamma_{12} = 4\zeta^2\omega_2(\omega_1 + \omega_2) - \varepsilon_{12} \tag{4}$$

$$\varepsilon_{12} =^2 \omega_2^2 - \omega_1^2 \tag{5}$$

$$k_{12} = \gamma_{12}\omega_1^2 + \gamma_{21}\omega_2^2 \tag{6}$$

where $\lambda_{i,U_j}$ is the spectral moment of order *i* of the response $U_j$ of the *j*-th frame (*j* = 1,2). Finally, Colajanni et al. [30] found that when the frames are excited beyond the elastic limit, the spectral moment should be evaluated for the hysteretic system according to the value

of the behavior factor that characterizes the ratio between the seismic input intensity and the strength of the frame. Alternative approaches able to take into account the effect of non-linear behavior on the effective properties of the frames, namely the increment of the period of vibration and the damping ratio, were developed by Penzien [41] or Jeng and Tzeng [17].

Concerning the problem of pounding, Jeng et al. [28] proposed the double difference modal combination rule, where in Equation (1) the value of $\rho_{12}$ is replaced with the value $\rho^{DD}_{12} = -\rho_{12}$ in Equation (2). Naderpour et al. [29], on the basis of the results of numerical analysis on MDOF systems and artificial neural networks, proposed to estimate the distance between two structures required to avoid pounding by using in Equation (2) the following combination coefficient:

$$\rho_{12} = 10.5(T_2 - T_1) - \frac{T_2}{T_1}; T_2 > T_1 \tag{7}$$

More recently, Miari and Jankowsky proposed new analytical expressions for providing a sufficient separation gap by considering the soil-structure interaction. The following values are suggested for structures founded on soil types A and B [24]:

$$\rho_{12} = \begin{cases} \left(\frac{T_1}{T_2}\right)^{-1.117} & T_1 \leq 0.2sec. \\ 57.343\left(\frac{T_1}{T_2}\right)^4 - 147.46\left(\frac{T_1}{T_2}\right)^3 + 141.74\left(\frac{T_1}{T_2}\right)^2 - 61.171\left(\frac{T_1}{T_2}\right) + 10.548 & T_1 > 0.2sec. \end{cases} \tag{8}$$

Referring to the NSA, researchers agree in the evaluation that basic procedures are suitable for capturing the response of the system vibrating according to the first mode only when the mass participation ratio in the analyzed direction exceeds 75%.

4.3.1. Modal Linear Dynamic Analysis in the Longitudinal Direction

Since the studied bridge is divided into two substructures by the constraints of the Gerber saddles, the results of standard Modal Dynamic Analysis (MDA) should be evaluated with special care. It has to be taken into account that since 1st mode and 3rd mode have similar periods, namely $T_1 = 1.74$ s and $T_3 = 1.67$ s, when the relative displacement between the two parts of the deck is evaluated, the Complete Quadratic Combination (CQC) rule instead of the standard SRSS rule should be used for the deck displacement at the abutment. By contrast, in order to evaluate the relative displacement at the RELC-LEIDIS, besides the SRSS and CQC rules, as well as the aforementioned DDC rule [28], the combination coefficient proposed by Naderpour et al. [29] (NAD), and proposed by Miari and Jankowsky [24] M&J should be considered.

In Figure 11b, the deformed shape due to the design seismic action ($PGA_d = 0.333$ g) acting in the longitudinal direction is shown, together with the zoom of the displacements at the left abutment (Figure 11a) and at the RELC-LEIDS (Figure 11c), where pounding or unseating are likely to occur. Table 1 gives the contribution of each mode in determining the displacement at the left and right abutment, and the relative displacement at the RELC-LEIDIS, with the value obtained by the modal combination rules. It has to be emphasized that the relative displacement should be evaluated by the modal combination of the relative displacement of each mode rather than between the difference of multimodal estimation of the maximum displacement of each node. This result can be obtained either by using correlation coefficients derived from noise modeling of the seismic excitation for an elastic system; alternatively, a more reliable estimation can be obtained by using a correlation coefficient that takes into account the energy content at the different frequencies of the input [39], or the expected non-linear behavior of the system [30]. In this regard, it is noteworthy that combination coefficients are estimated to be $\rho_{wn} = 0.960$ when white noise approximation of the seismic input is assumed (Equation (2)) and $\rho_{PSD} = 0.965$ when spectrum-compatible seismic input is assumed (Equation (3)). The coefficients suggested by Nadepur assume the value $\rho_{NAD} = 0.307$, while that proposed by Miari and Jankowsky

is $\rho_{M\&Y}$ = 0.691. Thus, the application of the SRSS provides an estimated displacement of 143 mm and 145 mm at the left and right abutments, respectively (Table 2). The relative displacement at the RELC-LEDIS is $\Delta d_{SRSS}$ = 207 mm, while the use of CQC, leaving the estimate of the displacement at the abutments almost unchanged, provides an increased value of the relative displacement, $\Delta d_{CQC}$ = 281 mm. Reduced values are provided by the modal combination rules derived on purpose for the pounding phenomenon: the DDC rule provides a very small value ($\Delta d_{DDC}$ = 79 mm); a value close to that provided by the SRSS rule is obtained by using the Naderpour coefficient ($\Delta d_{NDP}$ = 172 mm), and an intermediate value for the Miari and Jankowsky coefficient ($\Delta d_{NDP}$ = 115 mm).

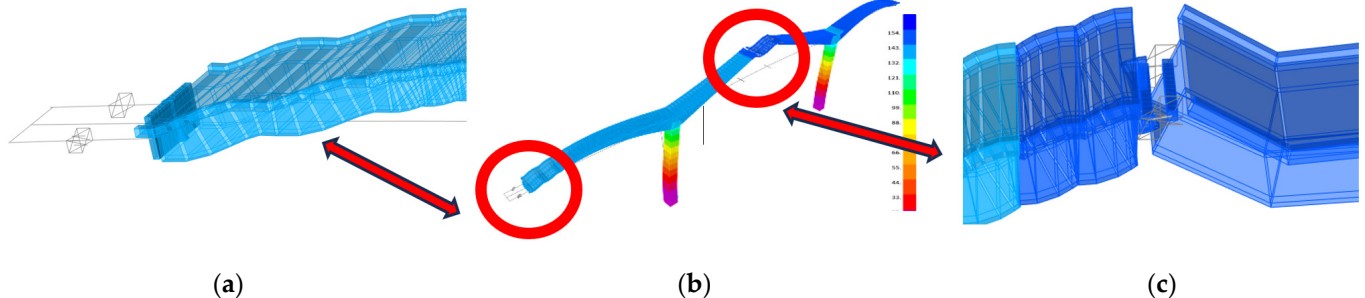

| (**a**) | (**b**) | (**c**) |

**Figure 11.** Longitudinal displacement obtained by linear modal dynamic analysis for PGA = 0.332 g. (**b**) deformed shape; zoom at: (**a**) the left abutment; (**c**) the joint between a deck of left cantilever and drop-in span.

**Table 2.** Assessment of deck displacement in the longitudinal direction at the abutments and RELC-LEDIS relative displacement by Modal Linear Dynamic Analysis for design value of $PGA_d$ = 0.333 g.

|  | Mode | | | | Modal Combination Rule | | | | |
|---|---|---|---|---|---|---|---|---|---|
|  | **1** | **3** | **5** | **6** | **SRSS** | **CQC** | **DDC** | **NAD** | **M&J** |
| LEFT ABUT. (mm) | 0.8 | −140.8 | −2.5 | 23.9 | 143 | 143 | | | |
| RIGHT ABUT. (mm) | 151.1 | −0.8 | 16.3 | 2.9 | 152 | 152 | | | |
| RELC-LEDIS (mm) | 151.9 | 140 | 18.8 | −21.0 | 207 | 281 | 79 | 172 | 115 |

Thus, while the modal dynamic analysis for the $PGA_D$ design value does not show a risk of pounding at the abutments, the risk of drop-in span unseating is estimated if the CQC rule is used for modal displacement superimposition.

### 4.3.2. Pushover Analysis in the Longitudinal Direction

Since the bridge is divided into two substructures by the constraints of the Gerber saddle between the left cantilever and the drop-in span, by performing push-over analysis with a load profile proportional to the product of the mass times a single-mode shape, one part of the bridge only is significantly loaded for each mode. Moreover, when each part is pushed in the positive or negative direction, the behavior can be very different due to the interaction of the deck with the abutments and at the RELC-LEDIS. Thus, in multimodal POA, when the modal contributions are combined, the sign of the applied forces and displacements (related to the sign of the modal shape) plays a role of great relevance. In fact, it determines the estimation of either pounding or unseating at the abutments and eventually, in the case of phase oppositions between the modes, either pounding or unseating at the cantilever deck drop-in span joint. Hence, the pushover analysis is able to determine the level of the seismic acceleration at which either pounding or unseating can occur at the abutments. By contrast, regarding pounding or unseating at the cantilever-drop-in span joint, only the boundary values of acceleration that define the potential occurrence of these phenomena can be assessed. This means that a lower bound corresponding to motions of the two parts that occur with a phase shift of 180° can

be obtained together with the upper bound corresponding to a null phase shift. Moreover, the single-run single-mode pushover response is the basis of the MPA.

It is noteworthy to recall that, as done in the previous section, in order to obtain the actual position of the bridge, the displacement due to dead load should be the initial displacement of the system at the beginning of the POA. In this case, in fact, a displacement of the node on the left cantilever of 43 mm is considered as well as a displacement of the right cantilever of −42 mm. Evaluation of the effects of creep and shrinkage, as well as those of temperature, are beyond the scope of the present research and thus were not considered.

4.3.3. Single-Mode Pushover Analysis

Figure 12a–d shows the deformed shape obtained with a load pattern consistent with a single-mode shape, while Figure 12e–h shows the corresponding pushover curves and the bilinear idealized curves. Table 3 gives, for each POA, the list of events with the corresponding values of the control node displacement and the corresponding level of PGA.

**Table 3.** Assessment of deck displacement in the longitudinal direction at the abutments and RELC-LEDIS relative displacement by Single-run single-mode POA for design value of $PGA_d$ = 0.333 g. PHA = plastic hinge activation; PHF= plastic hinge failure; YP = yielding of plug-in; FP = failure of plug-in; ES = exit from supports; L/R = left/right; PB = pier base; A = abutment.

| Mode | 1+ | | | 1− | | | 3+ | | | 3− | | |
|---|---|---|---|---|---|---|---|---|---|---|---|---|
| Event | | Displ (mm) | PGA (g) | Event | Displ (mm) | PGA (g) | Event | Displ (mm) | PGA (g) | Event | Displ (mm) | PGA (g) |
| ES DIS | 59 | 0.169 | | PHA RPB | −111 | 0.144 | PHA LPB | 110 | 0.097 | ES DIS | −59 | 0.170 |
| ES RA | 63 | 0.176 | | ES LA | −146 | 0.218 | ES LA | 148 | 0.152 | ES LA | −63 | 0.176 |
| PHA RPB | 74 | 0.194 | | POU DIS | −193 | 0.316 | POU DIS | 196 | 0.221 | PHA LPB | −76 | 0.199 |
| UNS DIS | 108 | 0.252 | | PHF RPB | −214 | 0.360 | PHF LPB | 217 | 0.251 | UNS DIS | −108 | 0.251 |

Assuming a load pattern according to the 1st mode in the +x direction ($f_{1+}$), only the right pier is significantly loaded (see Figure 12a), and thus the control node is taken at the left end of the deck; in Figure 12e, the pushover curve is depicted with the bi-linear equivalent curve corresponding to an equivalent SDOF with period $T_{1+}^* = 1.71$ s. A linear behavior is shown until the supports on the left end of the drop-in beam exit the bearing seat, and the friction starts to transfer part of the seismic load to the left pier. This phenomenon occurs for a seismic displacement of 101 mm, corresponding to a total displacement of $d_{+1,1}$ = 59 mm (Figure 12a and Table 3) and an associated $PGA_{+1,1}$ = 0.169 g. This event is followed by the same at the right abutment, which exits the seat for a seismic displacement of 105 mm (total displacement $d_{+1,2}$ = 63 mm and $PGA_{+1,2}$ = 0.176 g). The plastic hinge at the pier base is activated for a seismic displacement of 115 mm ($d_{+1,3}$ = 74 mm, $PGA_{+1,3}$ = 0.194 g). Unseating (downfall) at the left end of the drop-in occurs when the seismic displacement reaches 150 mm ($d_{+1,c}$ = 108 mm, $PGA_{+1,c}$ = 0.251 g), while the failure of the plastic hinge at the pier base occurs when the seismic displacement is 216 mm for a total displacement of $d_{+1,FPH}$ = 164 mm and $PGA_{+1,FPH}$ = 0.362 g.

Assuming a load pattern according to the 1st mode in the opposite direction (−x) that is ($f_{1−}$), the pushover curve, depicted in Figure 12b, is characterized by a trend with sections characterized by alternating increases and decreases in stiffness and a period of the equivalent SDOF of $T_{1−}^* = 1.91$ s. The first event is the activation of the plastic hinge at the base of the right pier for a seismic displacement of 69 mm (corresponding to a total displacement of $d_{1−,1}$ = −110 mm and $PGA_{1−,1}$ = 0.09 g); afterward, the deck at the right abutment exits, and the seat friction starts for a seismic displacement of −104 mm ($d_{1−,2}$ = −146 mm and $PGA_{1−,2}$ = 0.143 g). The closure of the gap between the deck and the left end of the drop-in span (pounding) occurs for a seismic displacement of 152 mm ($d_{1−,POU}$ = −193 mm and $PGA_{1−,POU}$ = 0.208 g). Lastly, the plastic hinge at the base of the right pier attains the ultimate rotation for a seismic displacement of −173 mm,

corresponding to a failure total displacement of $d_{1-,c} = -214$ and $PGA_{1-,c} = 0.360$ g for a smaller seismic displacement and a similar PGA value than those in the opposite direction.

The results for the 3rd mode, namely with load vector (f3) or (f3−), which produces the movement of the left pier, are similar to those of mode 1 and are not discussed for brevity's sake.

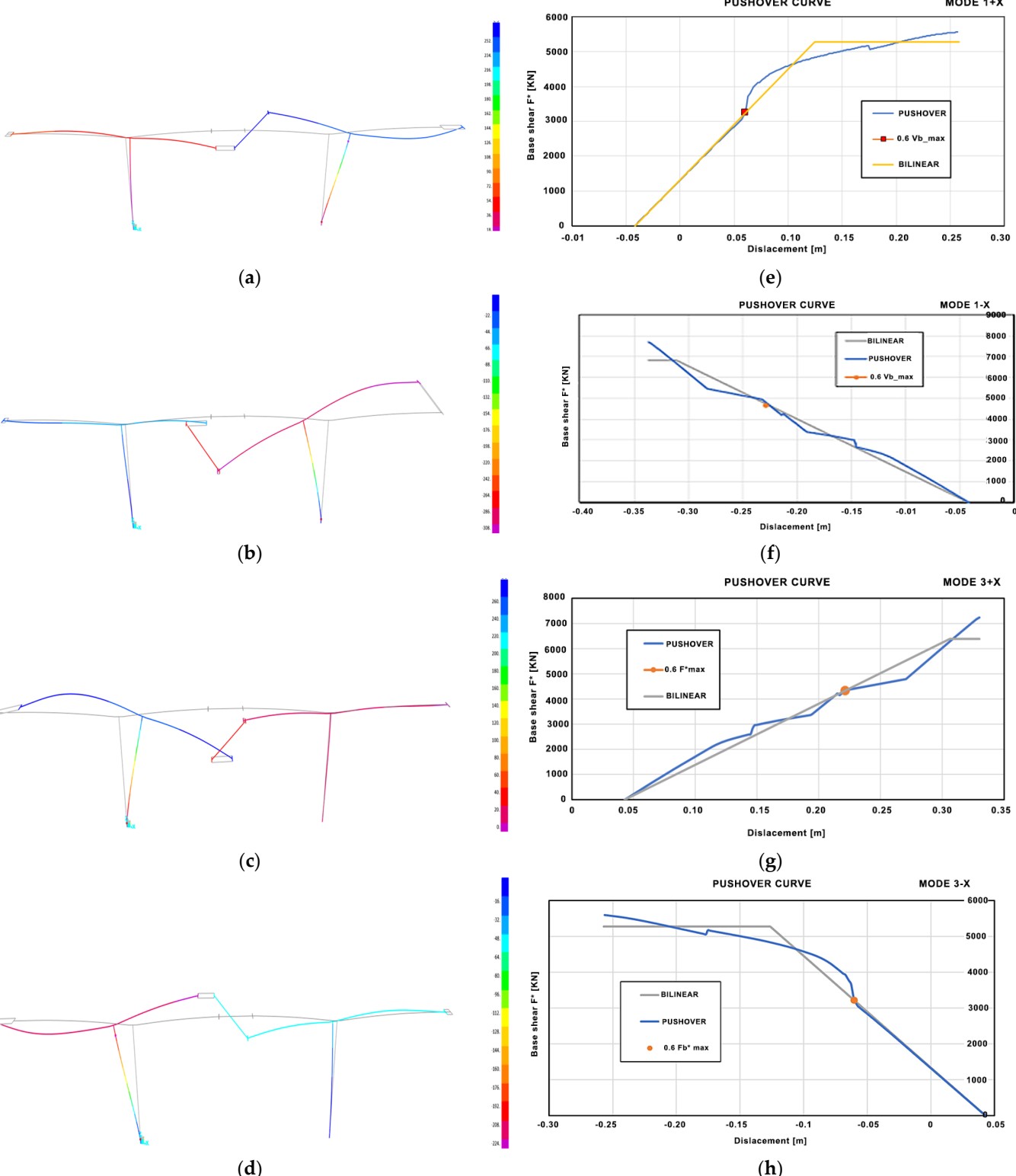

**Figure 12.** Single-mode POA: (**a**–**d**) deformed shapes; (**e**–**h**) pushover curves.

### 4.3.4. Single-Run Multi-Modal Pushover Analyses

The analyses presented in the previous section show that, when pushover analyses are performed for the bridge under examination, the combinations of modal contributions are not straightforward, either due to unsymmetric geometry or bilateral restraints. Moreover, the analysis performed in each mode is not able to completely reproduce the different interactions between structural elements and non-linear restraints.

Thus, in the following four additional single-run multi-modal analyses are performed; in each of them, the load pattern is obtained by the combination of the modal expansion of the load vectors according to the four permutations of the sign, namely $\pm(f1) \pm \alpha (f3)$, where $\alpha$ is the ratio of the elastic response spectrum for the period of vibration of the third and first modes, respectively. It must be noted that a linear combination can be assumed because the force vectors of the two modes act on almost mutually exclusive (different for each mode) nodes; in other ways, a modal combination rule is able to take into account the sign of each component also, should be applied. It is noteworthy that the signs in the above combinations are related to the phase shifting of the vibration of the two systems: if equal signs are assumed for the two "components" of the load vector, the absence of phase shifting ($0°$) is considered, while if opposite signs are assumed, a $180°$ phase shifting is considered.

Figure 13a–d shows the deformed shape obtained with the load pattern arising from the combination of first and third-mode shapes with the four permutations of the sign. Figure 13e,f shows the corresponding pushover curve for permutation characterized by concordant signs, namely $0°$ phase shifting, and the bilinear idealized curve is also depicted. It is noteworthy that, when opposite signs are considered in the combination, the total base shear is the difference between the absolute values of the base shear for each mode (Figure 13g,h), and it no longer represents the level of the seismic action. Thus, it is proposed to link the value of the PGA to the displacement of the control node of one part of the structure, as done in the single-run single-mode POA. In Table 4, for each of the aforementioned POAs, the list of events with the corresponding values of the control node displacement and the corresponding level of PGA is reported. For POA performed with $0°$ phase shifting, the sequence of the events and the corresponding displacement of the control node and PGA level resemble those obtained by "summation" of the two correspondent single-run single-mode POAs with the same sign in Section 4.3.3. The results prove that, as expected, when the vibration of the two substructures occurs with a small phase shift, their interaction is limited, and single-run single-mode POA is expected to be able to capture the actual structure behavior.

**Table 4.** Events that characterize the single-run multi-modal PO curve, with corresponding values of the displacement of the control node and related PGA.

| Dir x+ Phase Shift 0° | | | Dir x- Phase Shift 0° | | | Separ., Phase Shift 180° | | | Appr. Phase Shift 180° | | |
|---|---|---|---|---|---|---|---|---|---|---|---|
| Event | Displ (mm) | PGA (g) | Event | Displ (mm) | PGA (g) | Event | Displ (mm) | PGA (g) | Event | Displ (mm) | PGA (g) |
| PHA LPB | 112 | 0.111 | PHA RPB | 111 | 0.114 | ES DIS | 13 | 0.091 | PHA LPB | −88 | 0.064 |
| ES LA | 146 | 0.166 | ES LA | 131 | 0.146 | ES RA | 63 | 0.176 | POU DIS | −99 | 0.079 |
| ES RA | 149 | 0.171 | PHA LPB | 146 | 0.171 | PHA RPB | 75 | 0.196 | | | |
| PHA RPB | 214 | 0.274 | ES RA | 148 | 0.174 | ES LA | 81 | 0.206 | | | |
| PHF LPB | 214 | 0.275 | PHF LBP | 205 | 0.212 | UNS DIS | 109 | 0.254 | | | |

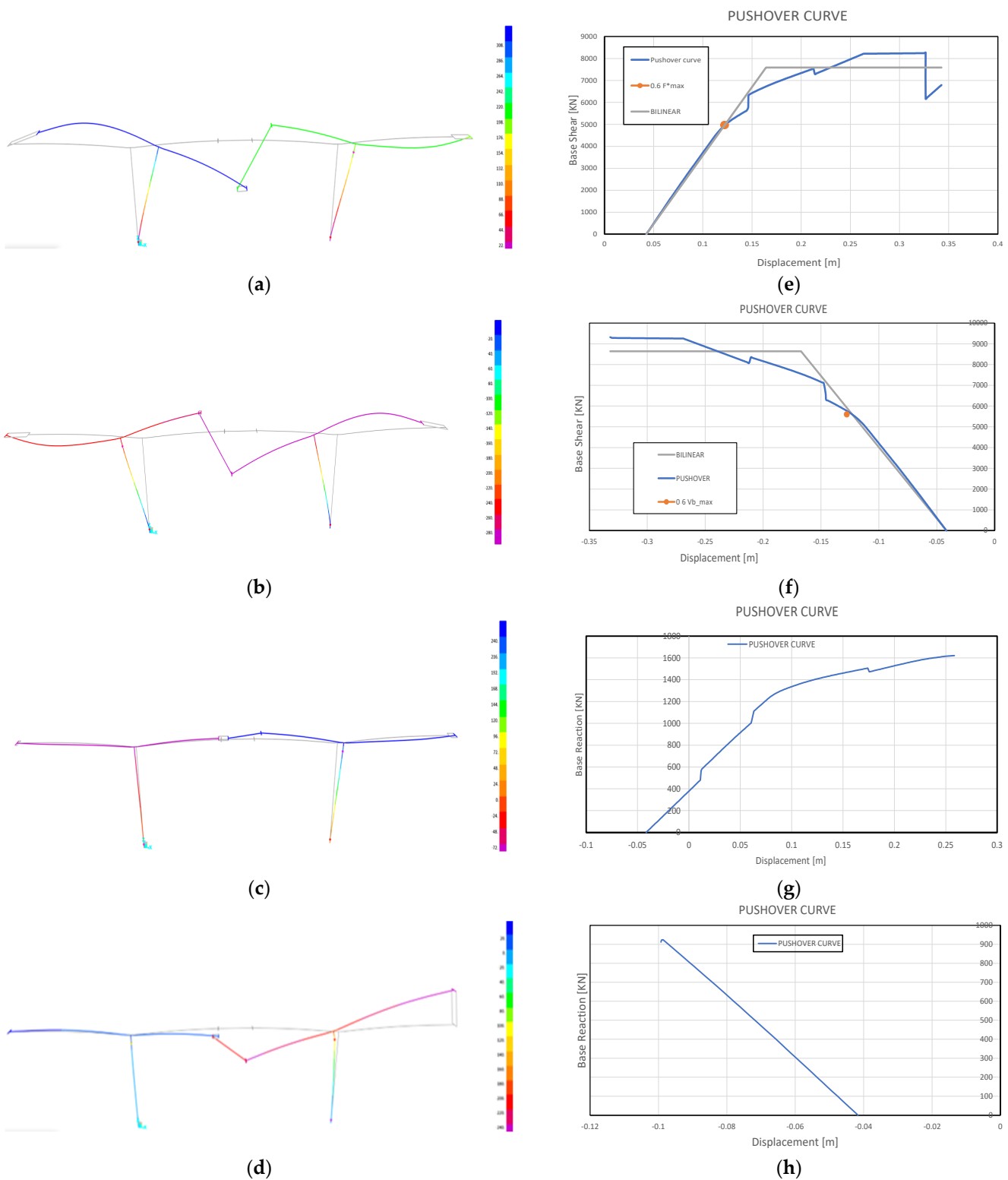

**Figure 13.** Single-run multi-modal modal POA: (**a**–**d**) deformed shapes; (**e**–**h**) pushover curves: (**a**–**e**) dir x+ phase shift. 0°; (**b**–**f**) dir x- phase shift. 0°; (**c**–**g**) separation, phase shift. +180; (**d**–**h**) approaching, phases shift. +180.

In Figure 13g, the pushover curve for the modal expansion of the load vector according to the combination $(f1) - \alpha(f3)$ that produces a separation of the two parts of the deck is represented, reporting on the vertical axis of the base shear obtained as the difference of the absolute values of the two parts of the load vector. As above-mentioned, in order to link the level of PGA to the displacement of the structure, the period of the SDOF equivalent system pertaining to the first mode only and the displacement of the corresponding control node are considered. The behavior is characterized (cfr. Table 4) by the exit of the supports of the RELC-LEDIS from its seat for a displacement due to seismic action of 54 mm (corresponding to a total displacement of $d_{(f1)-\alpha(f3),1} = 12$ mm and a $PGA_{(f1)-\alpha(f3),1} = 0.091$ g). This is followed by the exit of the supports at the right abutment for a seismic displacement of 104 mm (total displacement $d_{(f1)-\alpha(f3),2} = 63$ mm and $PGA_{(f1)-\alpha(f3),2} = 0.176$ g). Afterward, the plastic hinge at the base of the right pier is activated for a seismic displacement of 117 mm and $PGA_{(f1)-\alpha(f3),3} = 0.196$ g, followed by the exit of the supports at the left abutment for a seismic displacement of 123 mm and $PGA_{(f1)-\alpha(f3),3} = 0.206$ g. The collapse is due to the unseating of the RELC-LEDIS that occurs for a seismic displacement of 151 mm and $PGA_{(f1)-\alpha(f3),c} = 0.254$ g. The analysis was continued until a plastic hinge was activated at the base of the left pier, and the failure of the plastic hinge at the base of the right pier occurred for a seismic displacement of 217 mm and PGA = 0.347 g.

The pushover curve for the modal expansion of the load vector according to the combination $(f1) + \alpha(f3)$ that produces an approach of the two parts of the deck is represented in Figure 13h. The curve between the base shear (obtained as the difference of the absolute values of the base shear for each mode) and the displacement of the right end of the drop-in span (control node) is almost linear, thus the period of the SDOF equivalent system is taken as that of the 1st mode. The response is characterized by the activation of the plastic hinges at the base of the piers for a seismic displacement of the control node of $-46$ mm (corresponding total displacement $d_{-(f1)+\alpha(f3),1} = -88$ mm and $PGA_{-(f1)+\alpha(f3),1} = 0.082$ g), followed by the pounding of the RELC-LEDIS for a total displacement $d_{-(f1)+\alpha(f3),c} = -99$ mm and $PGA_{-(f1)+\alpha(f3),c} = 0.079$ g. These values represent the assessed pounding condition in the hypothesis of a phase shift of $180°$.

### 4.3.5. MPA Pushover Analysis

Due to the large stiffness of the deck in the longitudinal direction, the longitudinal displacements of the characteristic points are similar to each other; however, the participant mass ratio of the 5th mode (10%) is roughly one-third of that of the first mode (33%). Hence, by using MPA analysis, a larger displacement for a given PGA value is expected without relevant changes in the deformed shape. However, the non-linear analyses show an increment of 3.1% and 7.8% only for the displacement of the left and right parts of the deck, respectively. Regarding the relative displacement at the RELC-LEDIS, the MPA provides four different results, depending on how the modal contribution to the relative displacement is retained in the combination rule.

It is similar to what was highlighted in the previous section: to obtain a simultaneous approach or separation of the two ends or a simultaneous movement in the positive or negative direction of the axes. Moreover, the combination can be performed by using either the SRSS or the CQC rule for the deck displacement at the abutment; by contrast, to evaluate the relative displacement at the RELC-LEIDIS, besides the aforementioned combination rules, the combination coefficients described in Section 4.3 could be applied. The results obtained by using the SRSS and CQC rules are shown in Table 5. It must be emphasized that, since the two substructures have similar periods of vibration and a synchronous seismic excitation is assumed at the base of the piers, the actual expected displacement obtained by nonlinear time history analysis should resemble the results obtained with 0 phase shift; in this context, the role of CQC is of great importance in reducing the relative displacement expected at the RELC-LEIDS. The results obtained with a phase shift of $180°$ should represent the bound of the maximum relative displacement that can be expected when large non-linearities in the two systems can produce a large phase shift.

**Table 5.** Assessment of seismic displacement by MPA.

| | PH. SHIFT 0° + x | | PH. SHIFT 0° − x | | PH. SHIFT 180° APPR. | | PH. SHIFT 180° SEP. | |
|---|---|---|---|---|---|---|---|---|
| | SRSS | CQC | SRSS | CQC | SRSS | CQC | SRSS | CQC |
| LEFT ABUT. (mm) | 230 | 235 | 202 | 202 | 199 | 199 | 233 | 233 |
| RIGHT ABUT. (mm) | 214 | 250 | 190 | 193 | 199 | 190 | 206 | 126 |
| RELC-LEDIS (mm) | 243 | 171 | 241 | 216 | 268 | 293 | 213 | 237 |

### 4.4. Non-Linear Time History Analysis (NLTHA)

With the aim of evaluating the effectiveness of MDA and the pushover procedures described above, the results of non-linear time history analysis for the design value of $PGA_d = 0.33$ g are shown. Three accelerograms compatible with the elastic response spectrum for the $PGA_d$ value are generated according to the procedure described in [40], and they were made to act separately in both transverse and longitudinal directions.

In Figure 14, the transverse displacement along the longitudinal bridge axes obtained by single-run single-mode pushover and MPA are compared with the results of NLTHA.

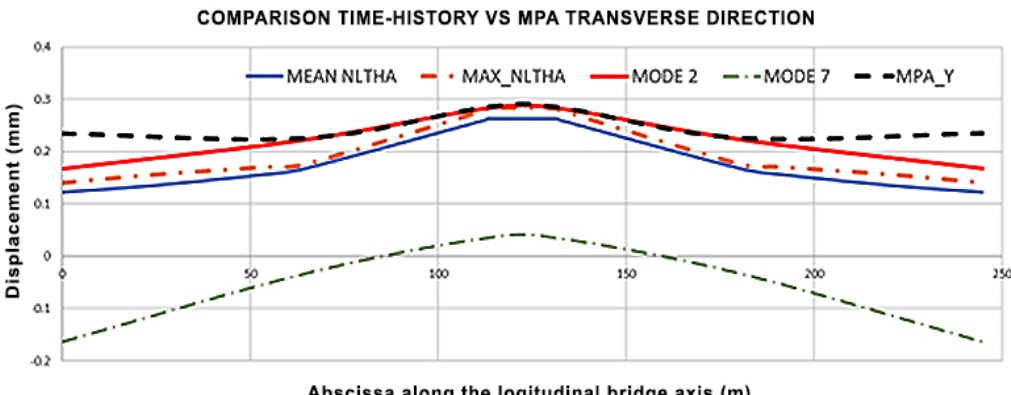

**Figure 14.** Comparison of assessment of transverse displacement along the bridge axis.

Preliminarily, it can be pointed out that the results of the maximum displacement at the midspan and the piers are slightly larger (roughly +6%) than the displacements caused by MDA, while since in MDA the abutments are restrained, the analysis method is not able to assess the actual response. Moreover, the figure shows that both the POA procedure predicts the same displacement of the control node (x = 101 m), equal to the maximum displacement obtained with the NLTHA, which must be taken as the design displacement in the case of using only three accelerograms. By contrast, on the piers (abscissa x = 51 m, and x = 171 m), and especially at the ends, MPA overestimates the displacement obtained by NLTHA, which is assessed with more accuracy by the simpler single-run analysis with a force profile proportional to the second mode only.

With reference to the response in the longitudinal direction, in Figure 15, the time history of the displacement obtained for the three NLTHA is compared with the displacement limit that produces the bearing exit from the seat, the unseating of all the deck ends, and the pounding of the deck with the abutment. The comparison with the results provided by MDA proves that linear analysis provides an underestimation of 27% of the maximum expected displacement. Moreover, the displacement estimation provided by the combination of the displacement obtained in single-mode (SM) pushover for 1st mode and 3rd mode with the hypothesis of synchronous motion is reported together with the results obtained by using single-run pushover analyses with load profiles obtained by the combination of first mode and third mode (PO SINCR MM), again with the hypothesis of synchronous motion. The results show that the latter is the procedure able to assess the actual displacement since it can consider the interaction between the different parts of the bridge.

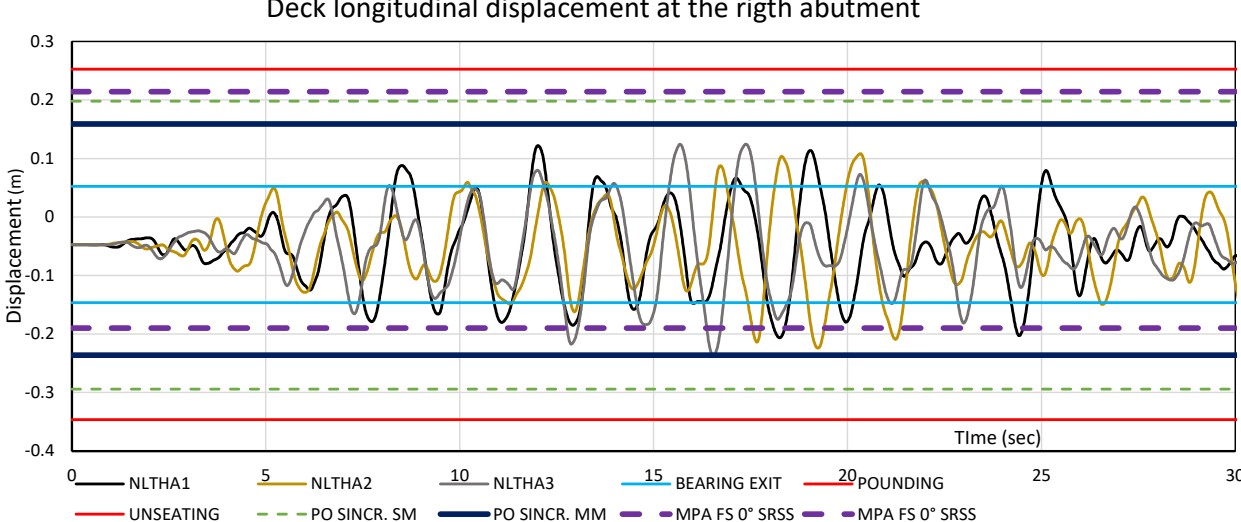

**Figure 15.** Comparison of longitudinal displacement assessment at the right abutment.

In Figure 16, the same comparison is shown for the relative distance of RELC-LEDIS due to the relative displacement, considering that the initial gap between the two ends after the effect of the gravitational load is 150 mm. The comparison with the results of MDA shown in Table 2, taking into account that the initial gap between the RELC-LEDIS is 150 mm; shows that only appropriate modal combination rules, e.g., DDC [28] and M&J [24], can provide a rough estimate of the actual response. Regarding POA methods, the results confirm that only PO SINCR MM can provide a reliable assessment of the response. The results of asynchronous motion, i.e., phase shifting of 180°, are not relevant in the studied case since the period of vibration of the two substructures are close to each other and the nonlinear behavior is also similar in the two substructures, at least until pounding or unseating occurs, a condition that lies beyond the field of this study. The latter results could be of practical interest when the periods of vibration of the two substructures are significantly different from each other or when asynchronous motions at the foot of the two substructures can occur.

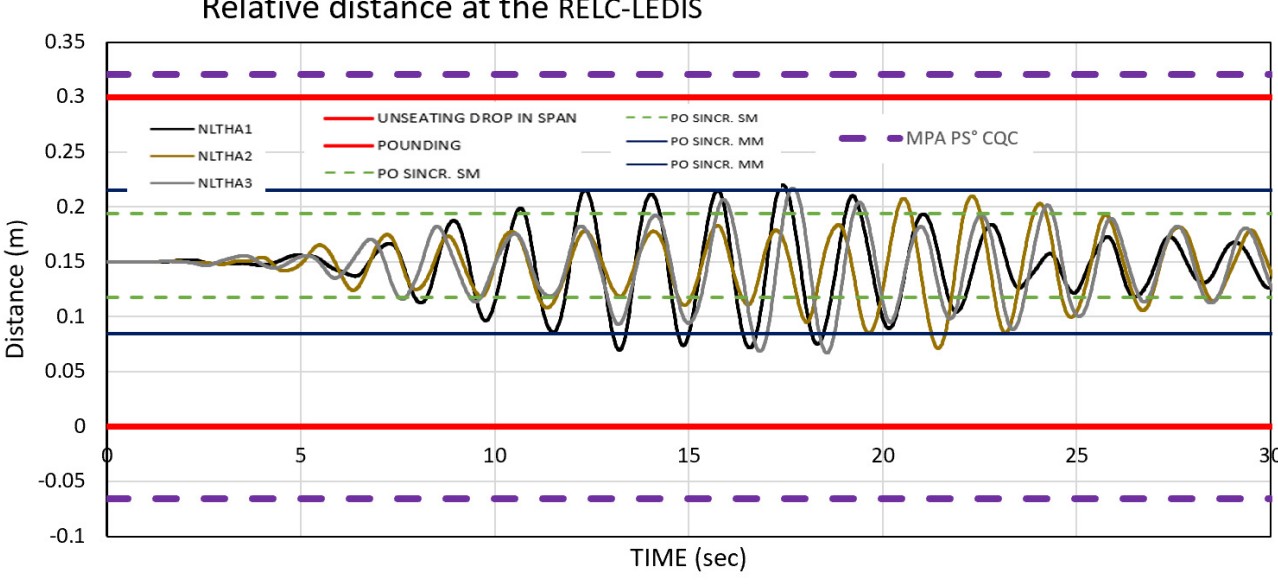

**Figure 16.** Comparison of relative displacement assessment of the left cantilever right end with respect to the left end of the drop in span.

## 5. Conclusions

A first attempt was made to highlight the problems in the use of pushover analysis for estimating the seismic vulnerability of bridges with reference to the service conditions and the ULS, due to the difficulties that emerged in the presence of drop-in spans and complex modeling of the nonlinear behavior of the bearing supports.

For the estimation of the behavior in the transverse direction, the difficulties that emerged in considering the higher modes were highlighted when these activate highly nonlinear behaviors (at the supports) for low levels of acceleration, making, in this case study, the multimodal techniques less accurate than the simple single-mode techniques.

With reference to the longitudinal direction, where pounding or unseating phenomena can occur, the need to resort to appropriate modal combination rules (which consider the contemporaneity of the signs of the different modes) was highlighted for the estimation of the response in the linear field.

In the nonlinear field, similar problems arise in the applications of the various PA techniques (created for the analysis of the behavior of buildings) in assessing the longitudinal behavior of bridges when risks of pounding or unseating phenomena occur.

For the case study, characterized by periods of vibration of the two substructures close to each other and similar nonlinear behavior, the best results have been obtained with single-run multi-modal pushover analysis techniques.

The results show that the role of the load pattern and the choice of a suitable multi-modal procedure, as well as detailed modeling of the bearings, are fundamental for the seismic vulnerability assessment both at the Serviceability Limit State (SLS) and Ultimate Limit State (ULS).

It has to be emphasized that only a case study was presented; thus the significance of the paper is limited to this kind of bridge. The paper highlighted some of the possible conditions in which the application of both modal dynamic analysis and pushover techniques to the assessment of the seismic vulnerability of bridges must be conducted with special care, through a critical interpretation of the modal combination rules to be applied in the first case and of pushover analysis techniques in the second.

In this connection, further studies are required to define a criterion for combining the modal contributions to the load vector when the presence of substructures characterized by different vibration periods and plasticization thresholds is considered, which should be validated by a more effective representation of the seismic input.

**Author Contributions:** Conceptualization, methodology, software, validation, writing—original draft preparation, P.C. and M.F.G.; writing—review, L.L.M.; funding acquisition L.L.M. All authors have read and agreed to the published version of the manuscript.

**Funding:** The studies presented here were carried out as part of both the activities envisaged by the Agreement between the High Council of Public Works (CSLLPP) and the ReLUIS Consortium implementing Ministerial Decree 578/2020 and Ministerial Decree 240/2022., and within the RETURN Extended Partnership and received funding from the European Union Next-GenerationEU (National Recovery and Resilience Plan—NRRP, Mission 4, Component 2, Investment 1.3—D.D. 1243 2/8/2022, PE0000005). The contents of this paper represent the authors' ideas and do not necessarily correspond to the official opinion and policies of CSLLPP.

**Data Availability Statement:** The data presented in this study are available on request from the corresponding author. The data are not publicly available due to rules of the founders.

**Acknowledgments:** We thank Ignazio Graceffo for the discussions on the topic and the collaboration in the development of some numerical analyses.

**Conflicts of Interest:** The authors declare no conflict of interest.

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
