# Peer review of "Seismic Vulnerability of Segmental Bridges with Drop-In Span by Pushover Analysis"

_applsci, doi:10.3390/app14010202_

Round 1

Reviewer 1 Report

Comments and Suggestions for Authors

The paper deals with an important and interesting topic in earthquake engineering. Pushover analysis was used to investigate the seismic vulnerability of segmental bridges with drop-in span in this paper. The topic of the paper falls in the scope of the journal. The method is sound and interesting, and the findings of this study are useful for earthquake engineers. Some comments from the reviewer are suggested below:

1.       Research significance of this study can be more clearly presented. Such as use the machine learning method can also help the seismic risk assessment of bridges (doi.org/10.1002/eqe.3699).

2.       Fig. 2 and Fig. 3 are not clear, more information can be provided, such as moment direction.

3.       More discussions of the results from push-over analysis and dynamic analysis can be provided.

4.       Fig. 5c is not clear. What are the values of the displacements?

5.       Limitations of the study should be presented.

Author Response

Please see the attachment. Moreover, a global revision of the language of the paper was made, trying to reduce the length and complexity of the sentences.

Reviewer 2 Report

Comments and Suggestions for Authors

This study evaluates the seismic vulnerability of bridges characterized by the presence of drop-in span by means of non-linear static analysis, with attention to the choice of the force distribution.

The outcomes show that, for bridge with the drop-in span, criteria for selecting the load pattern considered in pushover analysis, the reliable modelling of the bearings and tall piers play a dominant role in the assessment of the seismic vulnerability, particularly to longitudinal motion. It is a well written paper with detailed description of the pursued methodology and the produced results.

The paper could be accepted in the present form. 

Reviewer 3 Report

Comments and Suggestions for Authors

Lines 160-165 state specific results on the role of loading modes, the selection of multimodal procedures, and the detailed modeling of bearings in seismic susceptibility assessment that characterize the content belonging to the Abstract, Results, or Discussion sections. The introduction should generally avoid presenting specific findings; instead, it should summarize the research problem and objectives and set the context of the study.It would be more appropriate to reorganize this statement in the introduction by outlining the purpose of the research in this paper or the importance of these factors in seismic assessment, leaving the specific results to be discussed in detail in later sections.

Comments on the Quality of English Language
  1. Clarity and Conciseness: Some sentences are overly complex and could be simplified for clarity. For example, breaking down long sentences into shorter, more direct ones can enhance readability.
  2. Technical Jargon: While technical language is expected in a SCI paper, it should be balanced with clear explanations, especially when introducing less common concepts or methodologies.

Author Response

(The authors gave the same response as above.)
